# Genome-Wide Characteristics of GH9B Family Members in Melon and Their Expression Profiles under Exogenous Hormone and Far-Red Light Treatment during the Grafting Healing Process

**DOI:** 10.3390/ijms24098258

**Published:** 2023-05-04

**Authors:** Yulei Zhu, Jieying Guo, Fang Wu, Hanqi Yu, Jiahuan Min, Yingtong Zhao, Chuanqiang Xu

**Affiliations:** 1College of Horticulture, Shenyang Agricultural University, Shenyang 110866, China; zyl@stu.syau.edu.cn (Y.Z.);; 2Key Laboratory of Protected Horticulture (Ministry of Education), Shenyang 110866, China; 3Modern Protected Horticultural Engineering & Technology Center, Shenyang 110866, China; 4Key Laboratory of Horticultural Equipment (Ministry of Agriculture and Rural Affairs), Shenyang 110866, China

**Keywords:** β-1,4-glucanase, *CmGH9Bs*, graft healing, exogenous hormones, far-red light

## Abstract

β-1,4-glucanase can not only promote the wound healing of grafted seedlings but can also have a positive effect on a plant’s cell wall construction. As a critical gene of β-1,4-glucanase, *GH9B* is involved in cell wall remodeling and intercellular adhesion and plays a vital role in grafting healing. However, the *GH9B* family members have not yet been characterized for melons. In this study, 18 *CmGH9Bs* were identified from the melon genome, and these *CmGH9Bs* were located on 15 chromosomes. Our phylogenetic analysis of these *CmGH9B* genes and *GH9B* genes from other species divided them into three clusters. The gene structure and conserved functional domains of *CmGH9Bs* in different populations differed significantly. However, *CmGH9Bs* responded to cis elements such as low temperature, exogenous hormones, drought, and injury induction. The expression profiles of *CmGH9Bs* were different. During the graft healing process of the melon scion grafted onto the squash rootstock, both exogenous naphthyl acetic acid (NAA) and far-red light treatment significantly induced the upregulated expression of *CmGH9B14* related to the graft healing process. The results provided a technical possibility for managing the graft healing of melon grafted onto squash by regulating *CmGH9B14* expression.

## 1. Introduction

Grafting technology has been used as a conventional method in breeding fruit trees and vegetable seedlings [1,2], which can significantly improve a single variety’s yield, quality, and tolerance. It has been widely used in the cultivation and variety improvement of horticultural crops and created enormous economic value [3]. Due to the significance of grafted seedlings, there is a heightened curiosity in the grafting and recovery of grafted seedlings with a high survival rate. Melon (*Cucumis melo* L.) is an important economic fruit crop grown worldwide, with an annual output of more than 27 million tons in 2019 [4]. It originated in Africa and India [5] and is considered a rich source of multiple nutrients, such as vitamins, minerals, and dietary fiber [6]. Presently, melon grafting cultivation is a successful strategy for managing soil-borne diseases, overcoming the issue of continuous cropping, and boosting yield. *Cucurbita moschata* (*C. moschata*) is a genus of Cucurbitaceae squash that originated from Mexico to Central America, and it is widely cultivated around the world [7]. Grafting improved the absorption and utilization of water and nutrients due to the more developed root system [8], and grafted plants had resistance to a variety of soil-borne diseases. Therefore, squash is often used as the rootstock for grafting. However, in the actual production, due to the poor affinity between scion and rootstock, and long healing management cycle, the grafted seedlings have a low commercial rate and high seedling raising cost, which seriously affect the cultivation and production costs of grafted melon seedlings [9].

It was shown that the overexpression of β-1,4-glucanase promoted grafting [10], and a branch secreted into the extracellular region promoted cell wall reconstruction near the grafting interface. The *GH9* in plants has three branches (*GH9A*, *GH9B*, and *GH9C*) [11,12,13]. In avocado and pepper, the *GH9B* gene caused the fruit to stop growing and soften the tissue, which is related to the changes in the cell wall structure. In tomatoes, silencing of *SlGH9B2* increased the force required to separate the fruit from the plant [14]. In Arabidopsis, the *Cel3* and *Cel5* genes of *GH9Bs* were thought to promote cell wall loosening required for root cap cell shedding during root development and respond to biotic and abiotic rhizosphere stress. In poplar, the overexpression of *GH9Bs* reduced the amount of xylan cross-linked with cellulose microfibers and increased cell wall plasticity. The cellulose’s crystallinity and polymerization degrees were decreased by the overexpression of the *OsGH9B1* and *OsGH9B3* in rice, but the digestibility of cell wall enzymes was increased [15]. In poplar and Arabidopsis, the *GH9B5* gene was highly expressed during the development of the secondary wall, enhancing the cell wall structure [16]. During grafting between tobacco and Arabidopsis families, a large number of genes encoding cell wall modification enzymes were induced by transcription. Among them, the expression of the *GH9B3* gene encoding β-1,4-glucanase subclass increased continuously and promoted cell wall adhesion [17]. In tobacco, *NbGH9B3* was induced, along with other wall-remodeling genes at these graft junctions. The dysregulation of *NbGH9B3* significantly reduced the success rate of multiple homografts and heterografts [18]. In Solanaceae, the regulation mechanism of *GH9B3* gene expression has been changed. The expression of *GH9B* genes coding β-1,4-glucanase was relatively conserved in the grafting process of soybean and morning glories. The *GH9B3* gene improved the graft compatibility of petunias with different families and then enhanced the reproduction technology and flower yield [19]. During the interaction of parasitic plants, *S. japonicum* secreted the direct homolog of the *PjGH9B3* gene around the haustrum and directly contacted the host tissue, which reduced the thickness of the host wall at the interface between parasite and host plant cells and formed a xylem bridge to promote successful cell adhesion [20]. *GH9B* genes were demonstrated to be vital in the healing of grafted plants. The *GH9B* genes isolated and identified in Arabidopsis were divided into ten families (*GH9B1* family, *GH9B3* family, *GH9B7* family, *GH9B8* family, *GH9B10* family, *GH9B11* family, *GH9B12* family, *GH9B15* family, *GH9B16* family, and *GH9B18* family). Based on the phylogenetic classification of the *GH9B* family in Arabidopsis, the evolutionary relationships of the *GH9B* family members in other plant species can be analyzed using the same method. As plant genomic data continue to be published, members of the *GH9B* family will be identified from more and more plant species, which will lay the foundation for studying these genes from an evolutionary biological perspective.

The *GH9B* gene is essential for promoting graft healing and plant growth. However, the melon’s *GH9B* family members have not been thoroughly examined and characterized at the genomic level. We identified *GH9B* family members from melon genomes to improve the healing of the melon grafted onto the squash. This study aimed to investigate the structure and function of the *GH9B* gene in melon and the phylogenetic relationship between the melon protein and Arabidopsis thaliana. To determine the *CmGH9Bs* expression pattern, we deeply evaluated their responses to exogenous hormones and far-red light signaling. This study provided a theoretical basis for the functional analysis of *GH9B* family genes in melon that was of great importance for further regulating the healing of the melon grafted onto the squash.

## 2. Results

### 2.1. Identification of Members of the CmGH9B Gene Family Members

We identified 18 *CmGH9Bs* from the genome of melon genes and all *GH9B* genes mapped onto the melon chromosomes. According to the chromosome gene distribution order, they were named from *CmGH9B01* to *CmGH9B18*. The characteristics of *CmGH9Bs*, including the molecular weight, PI, instability index, and subcellular location of the 18 *CmGH9Bs*, were summarized in Table 1. The number of amino acids composing these *CmGH9Bs* varied greatly. The *CmGH9B1* comprised only 172 amino acids with a molecular mass range of 19,277.59 Da, and the *CmGH9B12* consisted of 722 amino acids with a molecular mass range of 79,876.41 Da. The *CmGH9B16* had the lowest PI (4.95), while the *CmGH9B11* had the highest PI (9.41). There were only nine genes, namely *CmGH9B1*, *CmGH9B4*, *CmGH9B6*, *CmGH9B7*, *CmGH9B8*, *CmGH9B12*, *CmGH9B16*, *CmGH9B17*, *CmGH9B18*, in the cell membrane; only two genes, namely *CmGH9B9* and *CmGH9B14*, in the cell wall; and seven genes, namely *CmGH9B2*, *CmGH9B3*, *CmGH9B5*, *CmGH9B10*, *CmGH9B11*, *CmGH9B13*, and *CmGH9B15*, in both the cell membrane and the cell wall (Table 1).

### 2.2. Phylogenetic Analysis, Classification, and Conserved Domain Analysis of CmGH9Bs

To study the phylogenetic relationships of 18 CmGH9Bs, we constructed phylogenetic trees based on amino acid sequence identity (Figure 1). The 18 CmGH9Bs were divided into three subtribes (I, II, and III), of which the largest group was Subtribe III (15 members), and Subtribes I and II contained one and two members, respectively (Figure 1A). Subtribe I included CmGH9B1, Subtribe II had CmGH9B7 and CmGH9B18, and the rest belonged to Subtribe III. The conserved domain analysis revealed that all CmGH9Bs contained the Clyco-Hydro-9 domain. However, not all CmGH9Bs had the CBM49 domain; it was mainly in Subtribe III (Figure 1B). Moreover, we represented the phylogenetic tree of relationships to analyze the evolutionary relationship between 18 CmGH9Bs and 10 AtGH9Bs. The results showed that these GH9B genes were divided into three subtribes (I–III). Subgroup III was the largest group, with 15 CmGH9Bs and 10 AtGH9Bs, the same subtribe and classification as CmGH9B alone (Figure 1C). Therefore, we speculated that CmGH9Bs and AtGH9Bs shared a common evolutionary origin, and CmGH9B14 was the homologous gene of AtGH9B3.

### 2.3. Motif and Exon–Intron Structure Analysis of CmGH9Bs

To determine the function of *CmGH9Bs*, the motif compositions of 18 *CmGH9Bs* were analyzed by the amino acid sequence in the MEME program. An analysis of 18 *CmGH9Bs* (Figure 2A) showed that, apart from *CmGH9B18* (no Motif2), *CmGH9B14* (no Motif3 or Motif5), and *CmGH9B1* (Subtribe I; no Motif1, -2, -3, -4, -5, -8, and -10), all the other motifs were present in the other 15 *CmGH9Bs.* These determine the classification, structure, and function of *CmGH9Bs*. Although their role has not been elucidated, they may indicate that the *GH9B* gene in melon has multiple functions. To identify the differences between the *GH9B* gene family in melon, we analyzed the *CmGH9Bs’* structure by comparing each coding sequence with the corresponding genome sequence. As shown in Figure 2, the numbers of *CmGH9Bs* exons were discontinuously and unevenly distributed from 1 to 18. Combining the gene structure and phylogenetic tree, we found that the exon–intron distribution of *CmGH9Bs* was related to their classification. Closely related genes usually had homologs. Therefore, their genetic structure was similar. For example, the *CmGH9B15* in the second subfamily had no introns, only exons. However, the *CmGH9Bs,* except for *CmGH9B15,* had exons and introns. Subgroup I had fewer exons (4 to 7) than other subgroups, and the exon–intron structural characteristics of Subgroups II and III were similar (Figure 2B). These results suggested that although *CmGH9Bs* were subdivided into three families, they were relatively genetically conserved.

### 2.4. Distribution of CmGH9Bs in Chromosomes

According to the melon genomic data, chromosome mapping of *CmGH9Bs* was carried out. The extraction of the chromosomal information of the *CmGH9Bs* identified their chromosomal locations. As shown in Figure 3, all *CmGH9Bs* had precise positions in the chromosomes. Each melon chromosome contains ≥1 *CmGH9B*. The *CmGH9Bs* were unevenly and non-randomly distributed on 15 chromosomes. Chr3 (chromosomal 3) had the four *CmGH9Bs*, and the rest of the Chr included only one *CmGH9B*. Chr3 was the longest in melon and consisted of many *CmGH9Bs*. The shortest chromosome, Chr14, had one *CmGH9B*. There was no significant correlation between chromosome length and *CmGH9B* gene distribution. There was no apparent correlation between the chromosome length and *CmGH9Bs’* distribution (Figure 3).

### 2.5. Cis-Acting Element Analysis of CmGH9Bs Promoters

To gain insight into the genetic functions, metabolic networks, and regulatory mechanisms of melon trihelix genes’ potential function of *CmGH9Bs*, the hypothetical promoters of cis-acting elements in 18 *CmGH9Bs* promoters (upstream 2 KB) were predicted and identified. The 18 *CmGH9Bs* had 627 cis-acting elements that reacted to the various abiotic stresses. Moreover, these cis-acting elements responded to gibberellin (GA), salicylic acid (SA), abscisic acid (ABA), methyl jasmonate (ME-JA), and so on. (Figure 4). Photoresponsive elements were detected in 18 melon mate promoters, accounting for the most significant proportion of all melon mate promoters. Among the 627 cis-acting factors, 19 cis-acting factors were associated with ME-JA response, distributed to 13 *CmGH9Bs*. The eight cis-acting factors were related to auxin response and distributed to six *CmGH9Bs*. The nine cis-acting factors were associated with GA response and distributed to seven *CmGH9Bs*. The seven cis-acting factors were related to SA response, allocated to six *CmGH9Bs*. Based on the above analysis, it was speculated that the *CmGH9Bs* played essential roles in response to the hormones.

### 2.6. Expression Patterns of the 18 CmGH9Bs in the Scion Stem during the Graft Healing Process

To better understand the expression pattern of 18 *CmGH9Bs* during graft healing, we measured and analyzed their relative expression in the melon scion stem of melon grafted onto squash rootstock by using qRT-PCR technology (Figure 5). The results showed that the relative expression level of *CmGH9B18*, belonging to Subtribe II, was the highest and significantly higher than the other 17 *CmGH9Bs*. In Subtribe III, the relative expression of *CmGH9B12* was markedly higher than the other 14 *CmGH9Bs*; the relative expression levels of *CmGH9B4*, *CmGH9B5*, *CmGH9B8*, *CmGH9B13*, *CmGH9B15*, and *CmGH9B17* were low, and there was no significant difference between them.

### 2.7. Effects of Exogenous Hormones on the Expression Pattern of 18 CmGH9Bs

Based on the cis-acting element analysis of 18 *CmGH9Bs* promoters, it was speculated that the *CmGH9Bs* played essential roles in response to the hormones. Thus, we analyzed the effects of exogenous NAA treatment on 18 *CmGH9Bs* expression in the scion stem of the melon grafted onto the squash rootstock (Figure 6). The results indicated the relative expression levels of six *CmGH9Bs*, namely *CmGH9B2*, *CmGH9B6*, *CmGH9B7*, *CmGH9B14*, *CmGH9B16*, and *CmGH9B17*, that were significantly upregulated; and the relative expression levels of two *CmGH9Bs*, namely *CmGH9B4* and *CmGH9B8*, that were significantly downregulated. There were no significant differences for the other 10 *CmGH9Bs* (*CmGH9B1*, *CmGH9B3*, *CmGH9B5*, *CmGH9B9*, *CmGH9B10*, *CmGH9B11*, *CmGH9B12*, *CmGH9B13*, *CmGH9B15*, and *CmGH9B18*) under exogenous NAA treatment. Notably, the relative expression level of *CmGH9B14*, a function analogous to *AtGH9B3* (Figure 1C), was significantly increased with the exogenous NAA application. Nevertheless, further research and evidence are necessary to ascertain whether NAA treatment can enhance the graft compatibility of melon scion and squash rootstock and promote its healing.

### 2.8. Effects of Exogenous Hormones and Far-Red Light Treatment on CmGH9B14 Expression during the Grafting Healing Process

To further verify the expression pattern of *CmGH9B14* during the graft healing process, we examined the relative expression of *CmGH9B14* with the NAA, SA, and far-red light treatment. In Figure 7, the relative expression levels of *CmGH9B14* during the graft healing process of melon scion grafted onto the squash rootstock were significantly higher than that of the own-rooted melon seedlings except for the 8 and 9 DAG. At the 1 DAG, the relative expression level of *CmGH9B14* was the highest. From 2 DAG to 9 DAG, the relative expression levels of *CmGH9B14* increased first and then decreased. Moreover, it maximized at 4 DAG and was significantly higher than the other days. It inferred *CmGH9B14* was very likely to be involved in regulating the grafting healing process of melon scion grafted onto the squash rootstock.

There were different expression patterns of *CmGH9B14* in the roots, stems, and leaves using the exogenous NAA and SA to treat the own-rooted seedlings of melon (Figure 8). The results showed that both NAA and SA significantly induced the upregulated expression of *CmGH9B14* in the melon roots (Figure 8A) and the downregulated expression of *CmGH9B14* in the melon leaves. Then, there was no difference between them. However, in the melon stems (Figure 8B), only the NAA treatment significantly increased the relative expression levels of *CmGH9B14*, and there was no difference between the SA and control. Therefore, we speculated that the exogenous NAA treatment accelerated the graft healing process of melon scion grafted onto the squash rootstock by inducing the upregulated expression of *CmGH9B14* in the stems of own-rooted melon seedlings.

To further determine whether far-red light affected the *CmGH9B14* expression in the grafting healing process, we treated the grafted melon seedlings with far-red light at 4 DAG. We measured the relative expression levels of *CmGH9B14* in the melon scion stems of grafted melon seedlings during the graft healing process. The results indicated that the relative expression levels of *CmGH9B14* at 6 DAG, 7 DAG, and 9 DAG after far-red light treatment were significantly higher than those of the control. Far-red light treatment could induce the upregulated expression of *CmGH9B14* and advance the expression peak of *CmGH9B14* during grafting healing (Figure 9).

## 3. Discussion

Plants are often affected by adverse environmental conditions throughout their life cycle, such as inadequate nutrient supply due to continuous cropping and invasion of various pathogens in the soil. These external environments can interrupt a series of critical physiological and biochemical processes in plants, resulting in growth arrest and even death. However, graft cultivation can significantly improve a single variety’s yield, quality, and tolerance. The results showed that the branch of β-1,4-glucanase promoted the cell wall reconstruction near the grafting interface and promoted grafting. The β-1,4-glucanase was found in prokaryotes and eukaryotes. It can efficiently degrade cellulose, complex insoluble substrates, and plant cell wall polysaccharides [21,22,23,24]. In addition, β-1,4-glucanase was involved in cell wall biosynthesis and modification, cell elongation and differentiation, cytokinesis, organ shedding, and fruit ripening [25,26,27,28]. Moreover, the β-1,4-glucanase was also involved in cell wall disintegration during leaf abscission [29]. In recent years, the *GH9B* gene family of β-1,4-glucanase has played a vital role in promoting graft healing [30] and has attracted more and more attention from researchers in basic and applied plant sciences. In this study, we conducted bioinformatics studies on the vital gene families in the grafting healing process of grafted seedlings and the effects of exogenous substances on family genes to observe the changes of the *GH9B* gene in the melon grafting healing process. We can conclude and propose that *CmGH9B14* may positively affect the grafting healing process of melon scion grafted onto squash rootstock.

### 3.1. Evolutionary Relationships and Characteristics of CmGH9Bs Family Members

*GH9B* belongs to the E2 subgroups of the glycosidic hydrolase family 9 (*GH9*) and has the capacity to secrete a single catalytic domain [31,32]. The *GH9B* genes were identified in Arabidopsis species for their powerful function in solving plant graft healing problems. Comparing the number of *GH9B* genes in Arabidopsis plants could better analyze the evolutionary relationship among *CmGH9Bs* and lay the foundation for graft healing. Currently, the *GH9B* genes reported in rice and Arabidopsis thaliana were most common in Arabidopsis thaliana, which could promote cell adhesion. However, it is not unknown about the information of *GH9B* genes in the Cucurbit melon.

In this study, 18 *CmGH9B* genes were identified based on the genomic data of melon. We also established phylogenetic trees to analyze the relationships among *GH9B* families in melon and Arabidopsis. The results showed that the *CmGH9Bs* were divided into three families (Figure 1). According to the classification method of Arabidopsis thaliana, *CmGH9B1* in melon belonged to Subtribe I, *CmGH9B7* and *CmGH9B18* belonged to Subtribe II, and Subtribe III contained 15 *CmGH9Bs*. Generally, *GH9B* from the same species is more closely related to other species. However, the *GH9B* genes of the same subfamily were more closely associated with the genes of other subfamilies, suggesting that these genes possibly come from the same ancestor. *CmGH9Bs* were unevenly located on the melon chromosomes, with chromosome 3 containing the most *CmGH9Bs* (Figure 3). According to phylogenetic tree, it seemed that the diversity in *GH9B* family members had probably occurred before divergences of monocot and dicot [33]. Based on the expression profile, *GH9B* genes showed diverse expression patterns suggesting that the mutation in regulatory sites of *GH9B* genes have affected the function and expression of *GH9B* family members [34,35]. In this study, *CmGH9Bs* belonged to 18 different response groups detected in the 2.0 KB promoter region of the melon with the help of the PlantCARE tool, among which hormones responding to *CmGH9Bs* included auxin, GA, ME-JA, and SA (Figure 4). Among the four types, ME-JA response elements accounted for the most significant proportion of melon promoters, followed by *CmGH9Bs* associated with auxin, GA, and SA. Cis-acting elements are essential for gene expression, and their number positively correlates with gene expression levels [36]. It is well-known that auxin is a crucial signaling molecule for most organogenesis and gene expression processes during plant growth and development [37]. Thus, we analyzed the effects of exogenous NAA treatment on 18 *CmGH9Bs* expression in the scion stem of the melon grafted onto the squash rootstock (Figure 6). The results also showed that the exogenous NAA treatment significantly increased the relative expression levels of *CmGH9B2*, *CmGH9B6*, *CmGH9B7*, *CmGH9B14*, *CmGH9B16,* and *CmGH9B17*.

### 3.2. CmGH9B14 Expression Profiles during the Graft Healing Process

Some reports have shown that the *GH9B3* encoding β-1,4-glucanase played an essential role in the graft healing of plants [17,18,19,20]. Through the evolutionary analysis of *CmGH9Bs* in melon, we found that *CmGH9B14* was the homologous gene of *AtGH9B3* (Figure 1C). In this study, we found that the relative expression levels of *CmGH9B14* increased first and then decreased during the graft healing process of melon scion grafted onto the squash rootstock (Figure 7). Furthermore, the *CmGH9B14* expression profiles were similar to *AtGH9B3, NbGH9B3,* and *SlGH9B3* [16,18,19], so we speculated that *CmGH9B14* was likely to regulate the grafting healing process of the melon scion grafted onto the squash rootstock. However, the molecular mechanism of *CmGH9B14* regulating the graft healing process of melon grafted onto squash needs further investigation. In addition, we also analyzed the *CmGH9B14* expression changes under the exogenous hormones (NAA and SA) and far-red light treatment during the graft healing process (Figure 8 and Figure 9). The results indicated that the exogenous NAA and far-red light treatment could significantly induce the upregulated expression of *CmGH9B14*. The results also provided a technical possibility for regulating the expression of *CmGH9B14* and then controlling the graft healing of melon grafted onto squash.

In summary, we identified 18 *CmGH9B* genes, respectively, based on the comprehensive analysis of melon genome data. Genetic information such as chromosomal location and exon–intron structure, conserved domains, and duplicated genes were provided, and the expression profiles of *CmGH9Bs* were explicitly examined. The *CmGH9B14* identified in this study can be used as a candidate gene to promote graft healing, but its specific functions need to be further verified by experiments. Screening candidate genes for graft healing can lay the foundation for improving the quality of grafted seedlings and provide valuable information for better production of grafted seedlings.

## 4. Materials and Methods

### 4.1. Identification and Sequence Analysis of CmGH9Bs Family in Melon

The melon *CmGH9Bs* were identified using a previously described method but with minor changes. The melon amino acid, genome, CDS sequence assembly, and corresponding annotation were downloaded from the Cucurbitaceae genome database on 15 December 2022 (CuGenDB, http://cucurbitgenomics.org/). The proteins of *CmGH9Bs* were verified using the Pfam and NCBI databases. Proteins obtained by the domain database screening confirmation were considered *CmGH9Bs* family members. The corresponding CDS and gene sequences were extracted according to their protein identifications. The MEME program (http://meme-suite.org/) [38] identified conserved motifs of the *CmGH9Bs* family proteins with the following parameters [39]. The structures of *CmGH9Bs* were displayed by the TBtools (https://github.com/CJ-Chen/TBtools) software. The chromosomal locations of *CmGH9Bs* were mapped onto the melon linkage map with an online tool, according to their TIGR numbers. Their molecular weight (Mw), isoelectric point (PI), the number of amino acids, instability index, and other information were subsequently obtained from ExPASy Website (http://www.expasy.org/). Use Euk-mPLoc2.0 (http://www.csbio.sjtu.edu.cn/bionif/euk-multi-2/Plant-mPLoc) to predict the subcellular localization information.

### 4.2. Phylogenetic Analysis

Multiple alignments were performed with the full-length amino acid sequences of *CmGH9Bs* proteins constructed by phylogenetic trees, using MEGA 7.0 (https://www.megasoftware.net/). Then, the *GH9B* protein sequence of Arabidopsis thaliana was obtained from the NCBI database, and the phylogenetic trees of melon and Arabidopsis thaliana were constructed similarly.

### 4.3. Chromosomal Mapping and Exon–Intron Distribution

Based on the initial position information of *CmGH9B*s in CuGenDB, the chromosomal location images of *CmGH9B* were drawn by TBtools [37]. The exon–intron distribution of *CmGH9Bs* was visualized with the help of TBtools (https://github.com/CJ-Chen/TBtools) software.

### 4.4. Cis-Element Analysis of CmGH9Bs

Promoters of the *CmGH9Bs* family genes were downloaded from the Ensembl Plants database (https://plants.ensembl.org/index.html). The online tools PlantCARE database (https://sogo.dna.affrc.go.jp/) and TBtools software were used to analyze the cis-regulatory elements of the *CmGH9B* family gene promoters [40].

### 4.5. Plant Growth Conditions and Treatments

The present study involved the application of the splicing graft method to graft the melon (HuaBao, *Cucumis melo var. Makuwa* Makino) onto the squash (ShengZhen No.1, *C. moschata*). This experiment was carried out during the scion’s first-true-leaf fully expanded stage and the rootstock’s cotyledon development stage, utilizing the one-cotyledon method described by Davis et al. [41]. When the scion was just exposed to the heart, we sprayed NAA solution (40 mg·L^−1^) and SA solution (100 mg·L^−1^) 4 times as treatment and sprayed distilled water as control. Grafted seedlings were transplanted in the nutritional bowl (12 cm × 12 cm) and moved into the healing chamber for grafted seedlings’ cultivation at Shenyang Agriculture University. At 4 DAG, the grafted seedlings were treated with far-red light for 110 min. The other management methods of grafted seedlings for the control, NAA, SA, and far-red light treatment were consistent [42].

### 4.6. Expression Analysis of CmGH9Bs

The grafting seedlings were divided into melon scion stem and squash rootstock stem. After sampling, the tissues were separated and immediately frozen with liquid nitrogen. After grinding the samples thoroughly in liquid nitrogen, the RNA of the samples was extracted by using an Ultrapure RNA Kit (Kangwei Century, Taizhou, China). An RNA reverse-transcription test was conducted, and the reaction system was as follows: 5x Prime Script RT Master Mix 5 µL, RNA 1000 ng, and ddH_2_O refill to 20 µL. The reaction procedure was 37 °C for 15 min, 85 °C for 5 s, and 4 °C for cooling. After the reaction, the obtained cDNA was diluted four times and then stored at −20 °C for subsequent qRT-PCR and gene cloning tests.

The transcriptional levels of different *CmGH9Bs* in three tissues of melon were measured using qRT-PCR, and the transcription levels of grafted seedlings with the exogenous hormones (NAA and SA) and far-red light treatment were also determined. The qRT-PCR primers of *CmGH9B*s were designed using Primer 5 (http://frodo.wi.mit.edu/).The internal reference gene uses β-actin. The qRT-PCR experiments were carried out via 2 × SYBR Green Pro Taq HS Premix (TaKaRa, Beijing, China). The gene expression data were calculated by the 2^−ΔΔ^CT method [43], the qRT-PCR experiments were carried out based on three biological replicates, three technical replicates were carried out for each biological replicate, and pictures were drawn using Prism 8.3.0. software.

### 4.7. Statistical Analysis

All experimental data, including those for correlation analyses, were processed using DPS statistical software V26.0 and expressed as the mean of three biological replicates of RNA sequence results or the mean ± standard deviation (SD) of three biological replicates. Differences between melon and squash samples were assessed at a significance level of 0.05 by one-way ANOVA tests.

## 5. Conclusions

In this study, the CmGH9Bs of melon were characterized at the genomic level. By studying the gene structure and motif composition of 18 CmGH9B genes, their evolutionary relationship with Arabidopsis GH9B genes, and their expression in different tissues of grafted seedlings, candidate genes that may promote graft healing were identified. In addition, the effects of exogenous hormones and far-red light treatment on CmGH9Bs expression profiles were preliminarily verified. Based on our analysis, we hypothesized that CmGH9B14 was likely to be a vital regulator of the graft healing process in melon scion grafted onto the squash rootstock. Although the functions of these candidate genes need to be confirmed, this study laid a solid foundation for promoting grafting healing and producing high-quality grafted seedlings.

## Figures and Tables

**Figure 1 ijms-24-08258-f001:**
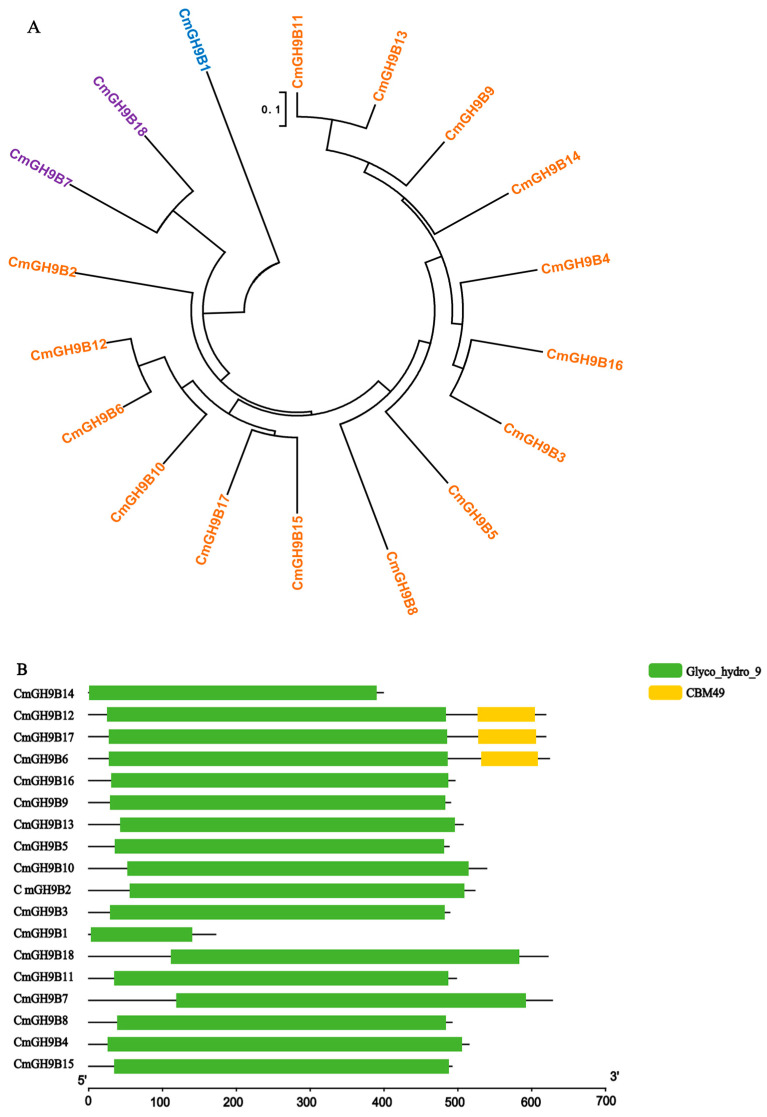
Evolutionary analysis of *CmGH9B*s in melon. (**A**) The *CmGH9Bs’* phylogenetic tree. (**B**) The *CmGH9Bs’* conserved domain. (**C**) The phylogenetic tree of relationships between 15 *CmGH9Bs* in melon and 10 *AtGH9Bs* in Arabidopsis. In the evolutionary tree, 0.1 represents the divergence point of a new gene replication or an organism that shares an ancestor, and the evolutionary distance between proteins is 0.1.

**Figure 2 ijms-24-08258-f002:**
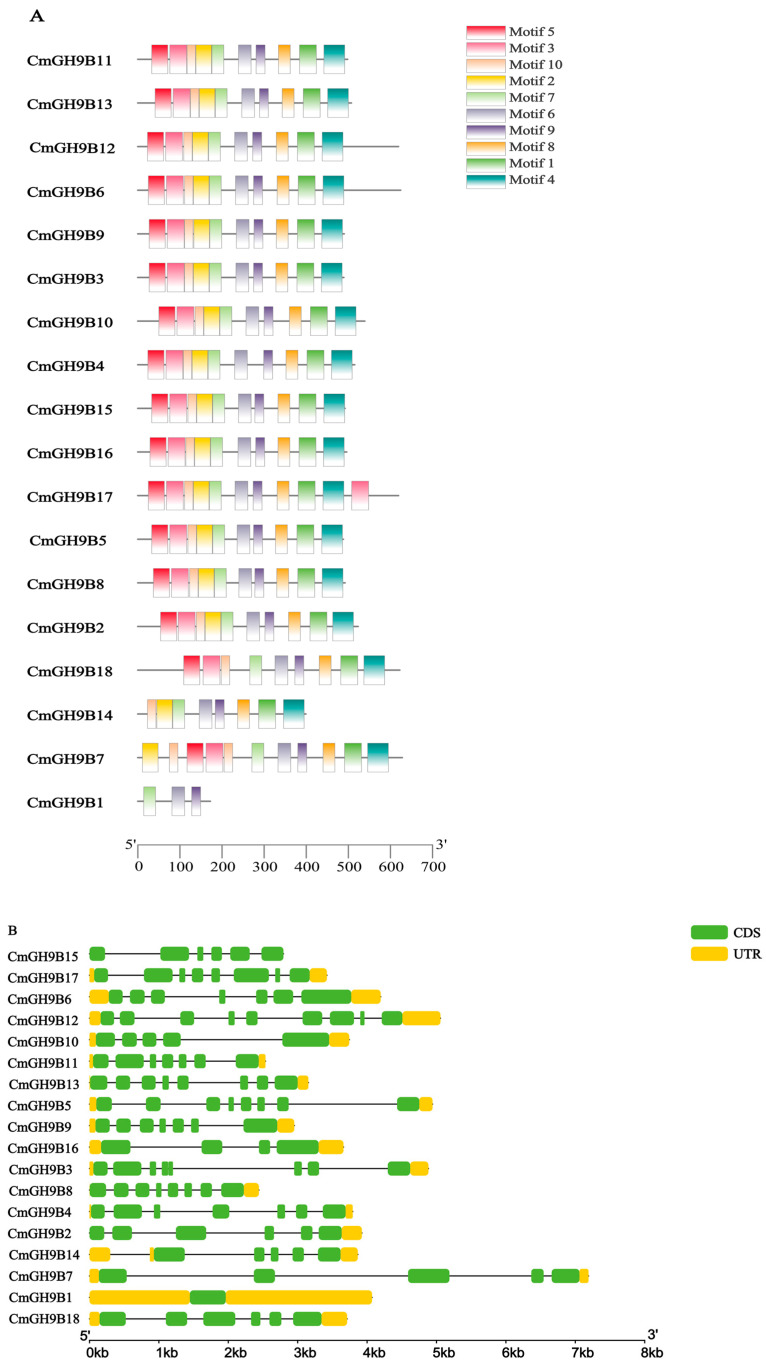
Motif and exon–intron structure analysis of *CmGH9Bs*. (**A**) Motif composition of *CmGH9Bs*. Boxes of different colors represented the various motifs. Their location in each sequence was marked. The scale bar at the bottom indicates the lengths of the *CmGH9Bs* sequences. (**B**) The exon–intron of *CmGH9Bs*. Round green rectangles, black lines, and yellow rectangles, respectively, marked the exons, introns, and untranslated regions. The scale bar at the bottom estimated the lengths of the exons, introns, and untranslated regions.

**Figure 3 ijms-24-08258-f003:**
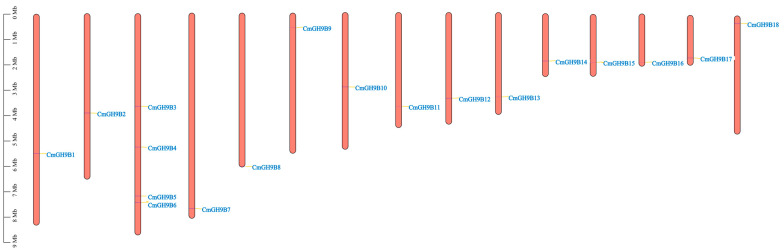
Chromosomal locations of *CmGH9Bs*. Orange bars represented the chromosomes. The different *CmGH9Bs* were labeled in green, at the right of the chromosomes. The scale bars on the left indicated the chromosome lengths (Mb).

**Figure 4 ijms-24-08258-f004:**
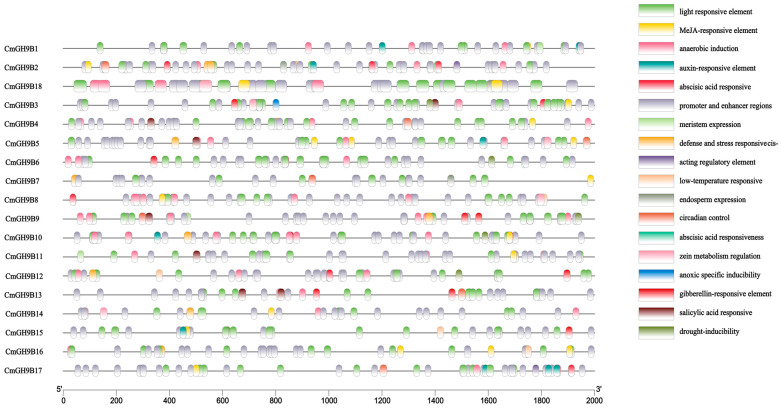
Predicted cis elements in the promoter regions of the *CmGH9Bs*. All the promoter sequences (−2000 bp) were analyzed.

**Figure 5 ijms-24-08258-f005:**
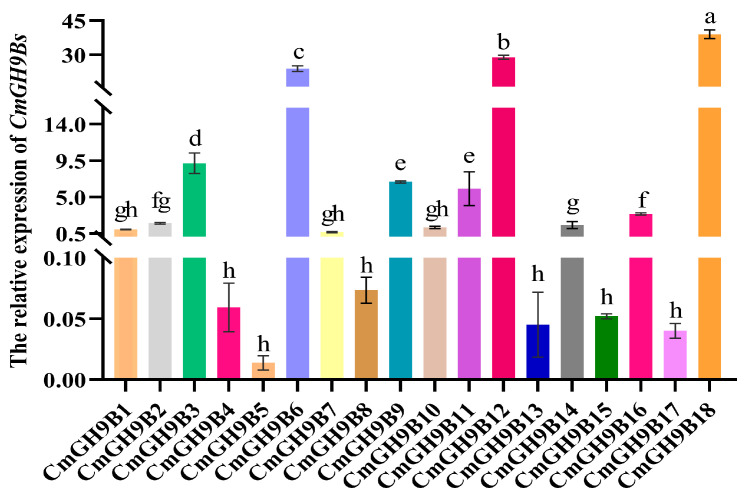
The expression pattern of 18 *CmGH9Bs* in the scion stem of melon scion grafted onto the squash rootstock. Different color columns indicate different *CmGH9Bs*. Different letters indicate significant differences (*p* < 0.05). Values are means ± SD, *n* = 3.

**Figure 6 ijms-24-08258-f006:**
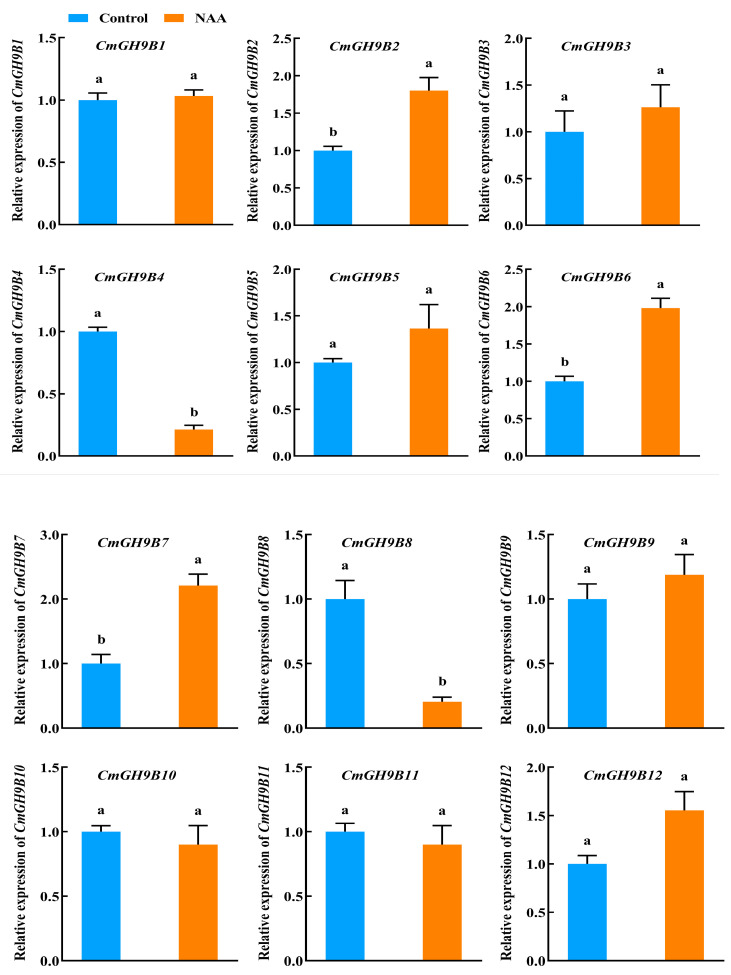
Effects of exogenous NAA on the relative expression levels of 18 *CmGH9Bs* in the scion stem of the melon grafted onto the squash rootstock. Different letters indicate significant differences (*p* < 0.05). Values are means ± SD, *n* = 3.

**Figure 7 ijms-24-08258-f007:**
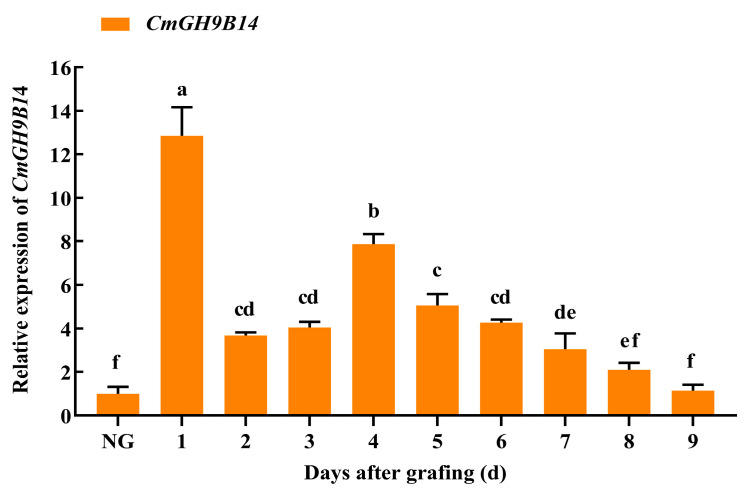
The expression pattern of *CmGH9B14* in the scion stem during the grafting healing process of melon scion grafted onto the squash rootstock. NG, no grafting. Different letters indicate significant differences (*p* < 0.05). Values are means ± SD, *n* = 3.

**Figure 8 ijms-24-08258-f008:**
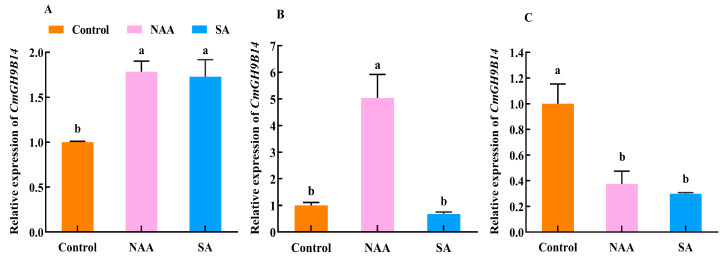
Effects of NAA and SA on the expression of *CmGH9B14* in the different tissues of the melon scion: (**A**) root, (**B**) stem, and (**C**) leaf. Different letters indicate significant differences (*p* < 0.05). Values are means ± SD, *n* = 3.

**Figure 9 ijms-24-08258-f009:**
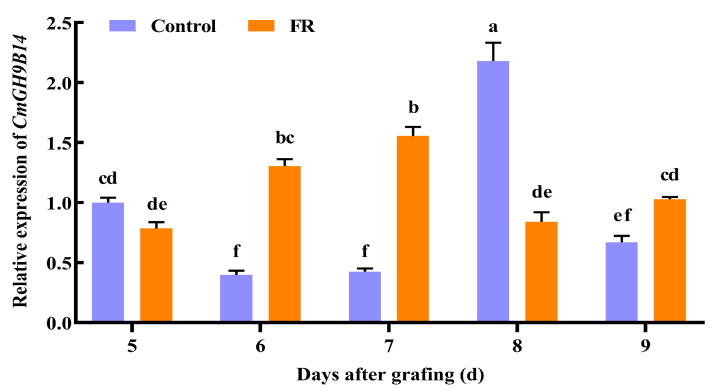
Effects of far-red light on *CmGH9B14* expression in the scion stem during the graft healing process of melon scion grafted onto the squash rootstock. FR, far-red light. Different letters indicate significant differences (*p* < 0.05). Values are means ± SD, *n* = 3.

**Table 1 ijms-24-08258-t001:** Physical and chemical characteristics of the 18 *CmGH9Bs* identified in the melon genome.

Gene Name	Gene ID	Instability Index	Number of Amino Acids	Molecular Weight/Da	PI	Molecular Formula	Subcellular Localization
*CmGH9B1*	LOC103489387	35.96	172	19,277.59	5.69	C_869_H_1308_N_224_O_264_S_5_	Cell membrane.
*CmGH9B2*	LOC103503885	34.84	525	57,454.67	5.45	C_2576_H_3922_N_666_O_787_S_20_	Cell membrane. Cell wall.
*CmGH9B3*	LOC103504547	46.42	280	31,269.31	7.05	C_1401_H_2142_N_396_O_405_S_8_	Cell membrane. Cell wall.
*CmGH9B4*	LOC103482534	38.98	430	47,711.83	8.50	C_2161_H_3226_N_558_O_632_S_18_	Cell membrane.
*CmGH9B5*	LOC103483071	39.40	488	54,715.70	8.41	C_2461_H_3722_N_652_O_729_S_19_	Cell membrane. Cell wall.
*CmGH9B6*	LOC103482819	38.84	567	63,475.65	8.75	C_2860_H_4296_N_766_O_831_S_24_	Cell membrane.
*CmGH9B7*	LOC103485910	32.02	628	69,807.10	9.37	C_3134_H_4772_N_862_O_905_S_25_	Cell membrane.
*CmGH9B8*	LOC103488526	47.37	526	58,743.14	8.45	C_2662_H_4068_N_692_O_765_S_23_	Cell membrane.
*CmGH9B9*	LOC103490702	41.16	490	53,778.91	5.36	C_2413_H_3637_N_629_O_746_S_12_	Cell wall.
*CmGH9B10*	LOC103491348	36.62	539	60,871.31	5.99	C_2731_H_4109_N_735_O_812_S_20_	Cell membrane. Cell wall.
*CmGH9B11*	LOC103492793	36.29	498	55,259.00	9.41	C_2486_H_3812_N_680_O_710_S_21_	Cell membrane. Cell wall.
*CmGH9B12*	LOC103493091	34.25	722	79,876.41	8.76	C_3638_H_5482_N_948_O_1048_S_19_	Cell membrane.
*CmGH9B13*	LOC103494756	34.42	507	55,924.32	8.65	C_2508_H_3851_N_675_O_743_S_18_	Cell membrane. Cell wall.
*CmGH9B14*	LOC103498660	35.08	494	54,386.60	8.98	C_2446_H_3731_N_659_O_715_S_18_	Cell wall.
*CmGH9B15*	LOC103499085	33.61	492	53,244.92	6.63	C_2389_H_3627_N_643_O_711_S_16_	Cell membrane. Cell wall.
*CmGH9B16*	LOC103499723	43.28	496	54,589.99	4.95	C_2448_H_3679_N_647_O_744_S_16_	Cell membrane.
*CmGH9B17*	LOC103499972	29.31	619	68,146.67	5.99	C_3089_H_4650_N_808_O_910_S_15_	Cell membrane.
*CmGH9B18*	LOC103493279	32.16	622	69,028.14	8.93	C_3121_H_4722_N_844_O_895_S_20_	Cell membrane.

## Data Availability

Not applicable.

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
