# Peer review of "Genome-Wide Characteristics of GH9B Family Members in Melon and Their Expression Profiles under Exogenous Hormone and Far-Red Light Treatment during the Grafting Healing Process"

_ijms, 2023, doi:10.3390/ijms24098258_

Round 1

Reviewer 1 Report

In the current study, GH9B family members in melon and their expression profiles under exogenous hormone and far-red light treatment during the grafting healing process were investigated. In my opinion, the novelty is low for publishing in IJMS. Also, the results are not well discussed. Other comments:

-          Line 41: Scientific name should be provided in italic. Please check the whole text.

-          Lines 58, 61, etc.: gene name must be in italic. Please check the whole text.

-          Line 99: Delete “genes” from “the genome of melon genes”

-          Lines 105-106: MWs are based on Da NOT KDa. Please correct them.

-          Discussion is so weak. This section is not acceptable.

-          Section 4.6 is not acceptable. More details of treatments such as NAA and SA application as well as far-red light treatment should be added.

-          I suggest transferring Table 2 to supplementary data.

Reviewer 2 Report

This study comprehensively characterized the GH9B family members in melon and investigated their expression profiles under hormone and far-red light treatment during the grafting healing process and highlight important roles of GH9B. There are some minor issues need to be addressed before further consideration.

Figure 1, the authors are suggested to add more species within Cucurbitaceae or from other families in the phylogenetic tree.

Figure 1B why GmGH9B1 has shorter domain compared to the others? Is this from the tech issue or the genome assembly issue?

Figure 1C The meaning of ‘0.1’ should be denoted in the legend.

Round 2

Reviewer 1 Report

The authors have managed to improve the article to some extent, but there are still some flaws. Apply the following comments:

-          Lines 499-500: Scientific names should be written in italic. Please check the whole text.

-          Discussion is still weak. I suggest adding these notes to Discussion: Line 510: According to phylogenetic tree, it seems that the diversity in GH9B family members have been probably occurred before divergences of monocot and dicot (Ahmadizadeh et al, 2020). Based on expression profile, GH9B genes showed diverse expression patterns suggesting that the mutation in regulatory sites of GH9B genes have affected the function and expression of GH9B family members (Yaghobi and Heidari, 2023; Hashemipetroudi et al, 2023).

-          Ahmadizadeh et al: https://doi.org/10.1016/j.genrep.2020.100894

-          Yaghobi and Heidari: https://doi.org/10.3390/genes14010202

-          Hashemipetroudi et al: https://doi.org/10.3389/fpls.2023.1112354
